

# A clustering method for small scRNA-seq data based on subspace and weighted distance

Zilan Ning[1,2], Zhijun Dai[1], Hongyan Zhang[2], Yuan Chen[1] and Zheming Yuan[1]

[1] Hunan Engineering & Technology Research Centre for Agricultural Big Data Analysis & Decision-Making, Hunan Agricultural University, Changsha, Hunan, China
[2] Hunan Agricultural University, College of Information and Intelligence, Changsha, Hunan, China

## ABSTRACT

**Background**. Identifying the cell types using unsupervised methods is essential for scRNA-seq research. However, conventional similarity measures introduce challenges to single-cell data clustering because of the high dimensional, high noise, and high dropout.

**Methods**. We proposed a clustering method for small **S**cRNA-seq data based on **S**ubspace and **W**eighted **D**istance (SSWD), which follows the assumption that the sets of gene subspace composed of similar density-distributing genes can better distinguish cell groups. To accurately capture the intrinsic relationship among cells or genes, a new distance metric that combines Euclidean and Pearson distance through a weighting strategy was proposed. The relative Calinski-Harabasz (CH) index was used to estimate the cluster numbers instead of the CH index because it is comparable across degrees of freedom.

**Results**. We compared SSWD with seven prevailing methods on eight publicly scRNA-seq datasets. The experimental results show that the SSWD has better clustering accuracy and the partitioning ability of cell groups. SSWD can be downloaded at https://github.com/ningzilan/SSWD.

## INTRODUCTION

Single-cell RNA-sequencing (scRNA-seq) technologies capture cellular heterogeneity between single cell, which allows researchers to dissect complex biological samples with detailed information about the transcriptome, thereby changing our understanding of biological systems (*Tang et al., 2009*; *Jaitin et al., 2014*; *Praktiknjo et al., 2020*). Identifying the cell types is essential in analyzing scRNA-seq data, and the quality will directly affect downstream analysis in single-cell (*Kharchenko, Silberstein & Scadden, 2014*). Unsupervised clustering is one of the most widely used methods for identifying cell groups in scRNA-seq data (*Ji & Ji, 2016*; *Žurauskiene & Yau, 2016*; *Kiselev, Andrew & Hemberg, 2019*; *Peyvandipour et al., 2020*; *Qi et al., 2020*). However, high dimensional, noise, and dropout characteristics of scRNA-seq data present traditional clustering methods with a

Corresponding authors
Yuan Chen, Chenyuan0510@126.com
Zheming Yuan, zhmyuan@sina.com

challenge (*Elowitz et al., 2002*; *Stegle, Teichman & Marioni, 2015*). Therefore, it is important to develop efficient and reliable clustering algorithms to identify cell groups.

Recently, many novel clustering methods have been developed for identifying cell groups of scRNA-seq data. Most of them focus on computing more accurate and robust similarity measures between cells (*Taiyun et al., 2018*; *Peng et al., 2020*). Single-cell Interpretation via Multi-kernel LeaRning (SIMLR) (*Wang et al., 2017*) chooses the most appropriate distance measure through multiple kernel learning and uses $k$-means to determine the cell groups. Seurat (*Satija et al., 2015*; *Butler et al., 2018*) and SNN-Cliq (*Xu & Su, 2015*) are graph-based clustering methods. Seurat constructs a $k$-nearest neighbor (KNN) graph with Euclidean distance in PCA (*Jolliffe, 2002*). SNN-Cliq combines a previously developed clustering algorithm with an SNN-based similarity measure, which determines cell groups automatically but requires three parameters to be specified. SC3 (*Kiselev et al., 2017*) employs consensus clustering to merge the clustering results under Euclidean distance, Pearson's correlation, and Superman's correlation to improve performance. However, SC3 is not scalable (*Kiselev, Andrew & Hemberg, 2019*). Besides, nonnegative matrix factorization, imputation, dimensionality reduction-based methods, and mixture model ensemble have been used to assess cellular heterogeneity (*Grün et al., 2015*; *Lin, Troup & Ho, 2017*; *Shao & Höfer, 2017*; *Yang et al., 2019*; *Huh et al., 2020*; *Venkatasubramanian et al., 2020*).

Subspace clustering is an efficient technique to mitigate noise applied in various fields (*Chen, Nasrabad & Tran, 2011*; *Ekström & Hagen, 2019*). SinNLRR (*Zheng et al., 2019*) considers cell clustering as a sparse subspace clustering (SSC) problem and uses the multiplier with an alternating direction to solve the optimization problem. S3C2 (*Zhuang et al., 2021*) combines enhanced SSC and low-rank completion algorithms in an optimization framework. DSCD (*Wang et al., 2020*) discovers the low dimensional latent structure from the compressed representation in scRNA-seq data and learns global relationships in single cells *via* a novel self-expressive denoise layer.

Highlighted by previous methods, calculating the similarity (distance) matrix of cells and reducing noise interference are crucial in clustering. This paper proposed a clustering method for small **S**cRNA-seq data based on **S**ubspace and **W**eighted **D**istance (SSWD), which assumed that sets of gene subspace composed of similar gene kernel density distributing genes could distinguish cell groups better. We proposed a new distance metric *EP_dis*, which integrates Euclidean and Pearson distance through a weighting strategy. Furthermore, we used the relative Calinski-Harabasz (RCH) index to determine the cluster numbers instead of CH because of its advantage of comparability in degrees of freedom. SSWD also included a consensus clustering process. Each of the gene subspace's clustering results was summarized using the consensus matrix integrated by PAM clustering. We applied the SSWD to eight public scRNA-seq datasets and contrasted it with seven widespread scRNA-seq clustering methods. The results show that SSWD reduces the influence of noise in clustering and better captures intrinsic relationships among cells or genes, which has greater clustering accuracy and the partitioning ability of cell groups.

## MATERIALS & METHODS

### Datasets

#### Simulated datasets

This paper used six simulation data to demonstrate the effect of *EP_dis* and RCH in the improved *k*-means algorithm. D1 and D2 were synthesized using different mathematical models (*Zhang, Yue & Zhang, 2014*) (Table 1). D1 contains five clusters with 420 (60, 80, 90, 90, 100) samples and 30 features. D2 contains four clusters with 300 (60, 70, 80, 90) samples and 10 features. Furthermore, four Gaussian datasets (D3-D6) (Fig. S1) (*Liu et al., 2010*; *Hussain & Haris, 2019*) were used to explain the properties of RCH with monotonicity, noise, density, and subcluster.

#### UCI datasets

Six real datasets (Table 2) from UCI (University of California Irvine) (https://archive.ics.uci.edu/) were used to validate the performance of RCH.

#### scRNA-seq datasets

We downloaded eight scRNA-seq datasets from GEO (https://www.ncbi.nlm.nih.gov/geo/) to validate the effectiveness of SSWD, for which the cell types were declared in the original publications. These datasets, including human and mouse species, involve various tissues and biological processes, such as cell development and differentiation, using different unit counts, *e.g.*, RPKM and FPKM. Specifically, *Biase, Cao & Sheng (2014)*, *Yan et al. (2013)*, and *Qiaolin et al. (2014)* consist of transcriptomes of human/mouse cells in embryos at some crucial developmental stages. *Treutlein et al. (2014)* contains 201 cells in four developmental stages of mouse lung epithelial cells. *Patel et al. (2014)* contains 430 glioblastoma cells from five patients. *Li et al. (2016)* is a human islet cell dataset, which contains alpha ($n = 18$), beta ($n = 12$), pp ($n = 9$), acinar ($n = 11$), and ductal ($n = 8$) cell subtypes. Tian307 and Tian305 (*Tian et al., 2019*) include lung adenocarcinoma cells from five patients. The detailed description of the datasets is listed in Table 3.

### The improved *k*-means algorithm with *EP_dis* and relative CH (RCH)

The *k*-means is a widely used clustering algorithm (*MacQueen, 1967*; *Jain, Murt & Flynn, 1999*; *Jain, 2008*). The algorithm requires the user to provide cluster initialization, distance metric, and the cluster numbers as the parameters (*Chiang & Mirkin, 2010*). Here we designed an improved *k*-means algorithm by introducing the *EP_dis* and RCH, which measure the similarity between two cells more appropriately and can automatically determine the cluster numbers.

#### EP_dis metric

Euclidean distance (E) is the most commonly used distance metric in traditional *k*-means, it characterizes the global correlation in high-dimensional space between samples. However, it will lose the correlation information between samples (cells or genes) when they have the same trend (*Taiyun et al., 2018*). Pearson distance (P) is another commonly used distance metric in clustering, which can captures the locally variable trend between samples (cells or genes), where P = (1 − R), and R is the Pearson correlation coefficient (*Fulekar, 2009*).

**Table 1** **Mathematical models of D1 and D2.** D1 contains five clusters with 420 (60, 80, 90, 90, 100) samples and 30 features. D2 contains four clusters with 300 (60, 70, 80, 90) samples and 10 features. $i, j$, and $k$ represent the cluster id, the feature id and the sample id, respectively. $\xi$: the random error.

| | D1 | D2 |
|---|---|---|
| cluster 1 | $0.1 + \sin(\frac{i}{3}) + \xi(i,j,k), \xi \sim N(0,1)$ | $\frac{-exp(j)}{1000} + \xi(i,j,k), \xi \sim N(0,1)$ |
| cluster 2 | $1.2\sin(\frac{2j}{5} - 2) + \xi(i,j,k), \xi \sim N(0,1)$ | $\frac{j}{6.6} + \xi(i,j,k), \xi \sim N(0,2)$ |
| cluster 3 | $1.5\sin(\frac{i}{3} - 3.5) + \xi(i,j,k), \xi \sim N(0,1)$ | $\frac{5(j-4)^2}{\max(i-4)^2} + \xi(i,j,k), \xi \sim N(0,2)$ |
| cluster 4 | $0.5\sin(\frac{2j}{5} - 2.2) + \xi(i,j,k), \xi \sim N(0,1)$ | $\sin(j) + \xi(i,j,k), \xi \sim N(0,1)$ |
| cluster 5 | $0.6\sin(\frac{i}{3} - 3.8) + \xi(i,j,k), \xi \sim N(0,1)$ | |

**Table 2** **Description of the six UCI datasets.** UCI (University of California Irvine) machine learning repository: https://archive.ics.uci.edu/.

| Datasets | No. of samples | No. of features | No. of categories |
|---|---|---|---|
| Dermatology | 366 | 33 | 6 |
| Seed | 569 | 7 | 3 |
| Sensor | 5,456 | 24 | 4 |
| Statlog | 2,000 | 36 | 6 |
| Waveform | 5,000 | 21 | 3 |
| Yeast | 1,484 | 8 | 10 |

**Table 3** **The details of eight scRNA-seq datasets.**

| Datasets | Groups | Variables | Cells | Units | Species | Protocol | Reference |
|---|---|---|---|---|---|---|---|
| Biase | 3 | 25737 | 49 | FPKM | Mus musculus | Smart-Seq | *Biase, Cao & Sheng (2014)* |
| Li | 5 | 180253 | 58 | RPKM | Homo sapiens | Smart-Seq2 | *Li et al. (2016)* |
| Patel | 5 | 5948 | 430 | TPM | Homo sapiens | Smart-Seq | *Patel et al. (2014)* |
| Deng | 7 | 12735 | 135 | RPKM | Mus musculus | Smart-Seq2 | *Qiaolin et al. (2014)* |
| Treutlein | 4 | 11245 | 201 | FPKM | Mus musculus | SMARTer | *Treutlein et al. (2014)* |
| Yan | 7 | 12325 | 90 | FPKM | Homo sapiens | Smart-Seq2 | *Yan et al. (2013)* |
| Tian307 | 5 | 13800 | 307 | UMI | Homo sapiens | CEL-Seq2 | *Tian et al. (2019)* |
| Tian305 | 5 | 13137 | 305 | UMI | Homo sapiens | CEL-Seq2 | *Tian et al. (2019)* |

**Notes.**
FPKM, fragments per kilobase of transcript per million mapped reads; RPKM, reads per kilobase of transcript per million mapped reads; TPM, transcripts per million mapped reads; UMI, unique molecular identifiers.

Here, we combined Euclidean and Pearson distances through a weighting strategy and defined a new distance, *EP_dis* metric (*Ning et al., 2022*). It was defined as follows:

$$EP\_dis = wE + (1-w)P. \tag{1}$$

A bigger *EP_dis* shows a weaker similarity between samples. If $w = 0$, *EP_dis* is Pearson distance; if $w = 1$, it is Euclidean distance. $w$ is the weight, and it ranges from 0 to 1. The matrix $E$ and $P$ must be min-max normalized when calculating *EP_dis* because the range of $E$ and $P$ are different. Take the maximum $SS_B/SS_W$ as the standard, and a step-by-step search determines the suitable $w$ in *EP_dis*. Where $SS_B = \sum_{i=1}^{k} n_i ||c_i - \bar{c}||^2$ represents the sum of squares between clusters and $SS_W = \sum_{i=1}^{k} \sum_{j=1}^{n_j} ||x_j - c_i||^2$ represents the sum of

squares within clusters; $k$ represents the cluster numbers; $n_i$ ($n_j$) represents the sample numbers in cluster $V_i$ ($V_j$); $\bar{c} = \sum_{i=1}^{N} \frac{x_i}{N}$ is the overall mean; $N$ is the sample numbers. We adopt the maximum technique (*Fränti & Sieranoja, 2019*) to obtain the cluster's initial centroids to ensure clustering stability.

### Determine the number of clusters

The $k$-means algorithm needs to be specified the number of clusters. The clustering internal validation (CIV) indices, such as the Calinski-Harabasz (CH) index (*Caliński & Harabasz, 1974*), Silhouette (*Sil*) index (*Rousseeuw, 1987*), and *Gap* Statistic (*Tibshirani & Hastie, 2001*), can be used for estimating the cluster numbers. The CH has been proven the best in estimating cluster numbers (*Milligan & Cooper, 1985*; *Chiang & Mirkin, 2010*). It is defined as:

$$CH = \frac{\frac{SS_B}{k-1}}{\frac{SS_W}{N-k}}, k = 2, 3 \dots NC, \tag{2}$$

where $N$ as the sample numbers, $NC$ as the largest cluster numbers. The $k$ with the maximum CH is the suitable cluster numbers. In different $k$, the CH value is incomparable because the degrees of freedom vary. So, we designed a new index, relative CH (RCH) (*Ning et al., 2022*), that was relatively comparable under different $k$:

$$RCH_k = \frac{CH_k}{F_{(\alpha, k-1, N-k)}}. \tag{3}$$

The workflow of the improved $k$-means algorithm with $EP\_dis$ and RCH is shown in Fig. 1.

## The overview of the SSWD

In the scRNA-seq data matrix $X_{G \times N} = \{x_{ij} | 1 \le i \le G, 1 \le j \le N\}$, rows represent genes, and columns represent cells. $x_{ij}$ represents the value of gene $i$ in the $j$ th cell. The framework of SSWD is depicted in Fig. 2.

### Step 1 filtering genes

Since rare and ubiquitous genes provide insufficient information for clustering, we only retained the $v$ genes (default: 1,000) with the highest variance after log-transformed. Specifically when the maximum value in $X$ is greater than 10,000, $X' = \log10(X + 1)$, otherwise $X' = \log2(X + 1)$. The gene subset $X'$ was the input of the second module.

### Step 2 partition genes with subspace

In scRNA-seq data, the sets of subspace represent the groups of genes. The gene subspace with similar density genes can distinguish informative features from noise (*Song et al., 2021*). We used the function *density* in R to calculate the gene's density. Specifically, the kernel density function scattered the density of genes over a regular grid of 512 points and convolved this approximation with the discretized kernel version using a fast Fourier transform. Then the function used the linear approximation to evaluate the density at each point (*Sheather & Jones, 1991*). In the density matrix $E_{G' \times 512}$, column and row represent the density values and gene, respectively. The improved $k$-means algorithm with $EP\_dis$ and RCH was employed to group the genes with similar density in matrix '$E$'. Then, the

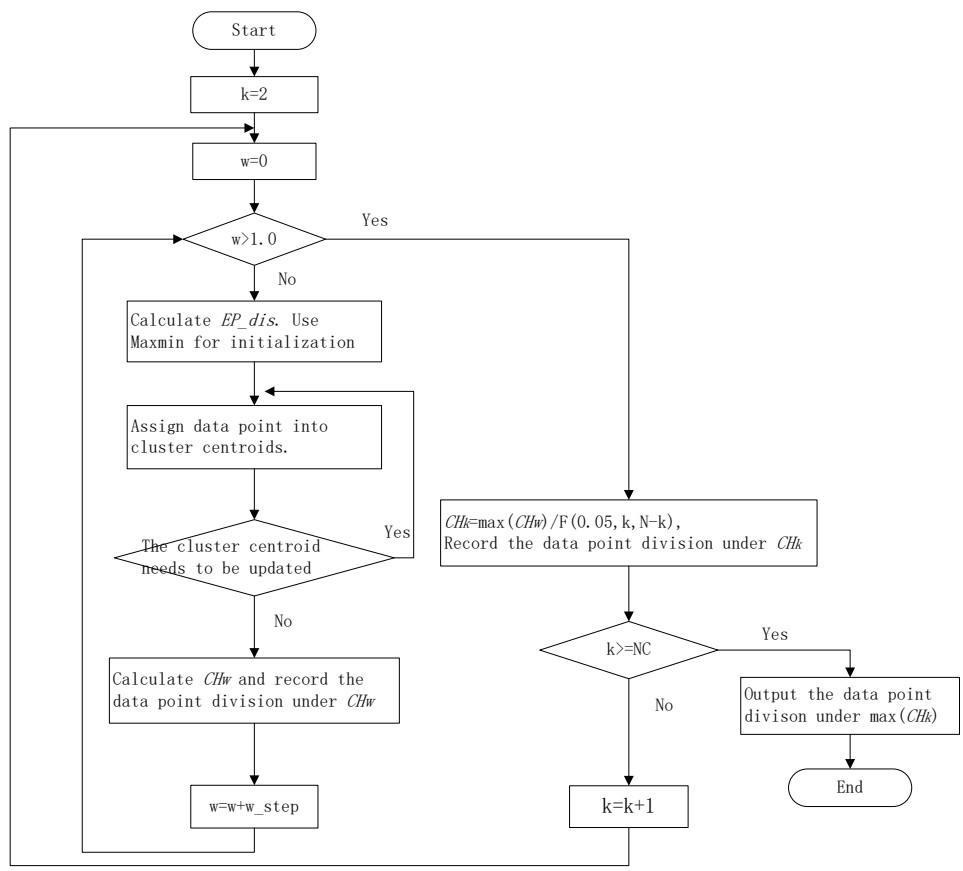

**Figure 1 The workflow of improved *k*-means algorithm with *EP_dis* and relative CH.** NC is the largest cluster numbers; *N* is the sample numbers; *w_step* is the search step; *F* (0.05, *N*, *N-k*) is the corresponding *F*-test threshold at the significance level of 0.05.

$X'_{G' \times N}$ was separated into several sets of gene subspace. Each set of gene subspaces contains all cells and some genes $X'_{G' \times N} = \bigcup_{i=1,2,...,c} subspace_i$, $subspace_i = X'_{N \times G'_i}$.

### Step 3 cell clustering in subspace

The sets of gene subspace containing more than three genes have been kept. Then, we used PCA for dimensionality reduction and retained the first *d*-dimension with the Elbow method (*Thorndike, 1953*). Then the improved *k*-means algorithm with *EP_dis* and RCH was employed to get the sets of gene subspace clustering results $Y_{subspace_i}$.

### Step 4 consensus clustering

The cluster-based similarity partitioning algorithm (CSPA) was used to compute the consensus matrix *M* (*Strehl & Ghosh, 2002*). $M_{N \times N} = \{M_{ij} | M_{ij} = num\}$, $(i, j = 1, 2, ...N)$, based on the clustering results from the sets of gene subspace. The *num* is the number of subspaces where cells *i* and *j* are in the same cluster. If *num* =0, cell *i* and *j* are never in the same subspace. Because the *M* was a discrete matrix, the improved *k*-means with *EP_dis* and RCH were unsuitable, but the PAM algorithm did. PAM is a variation of the

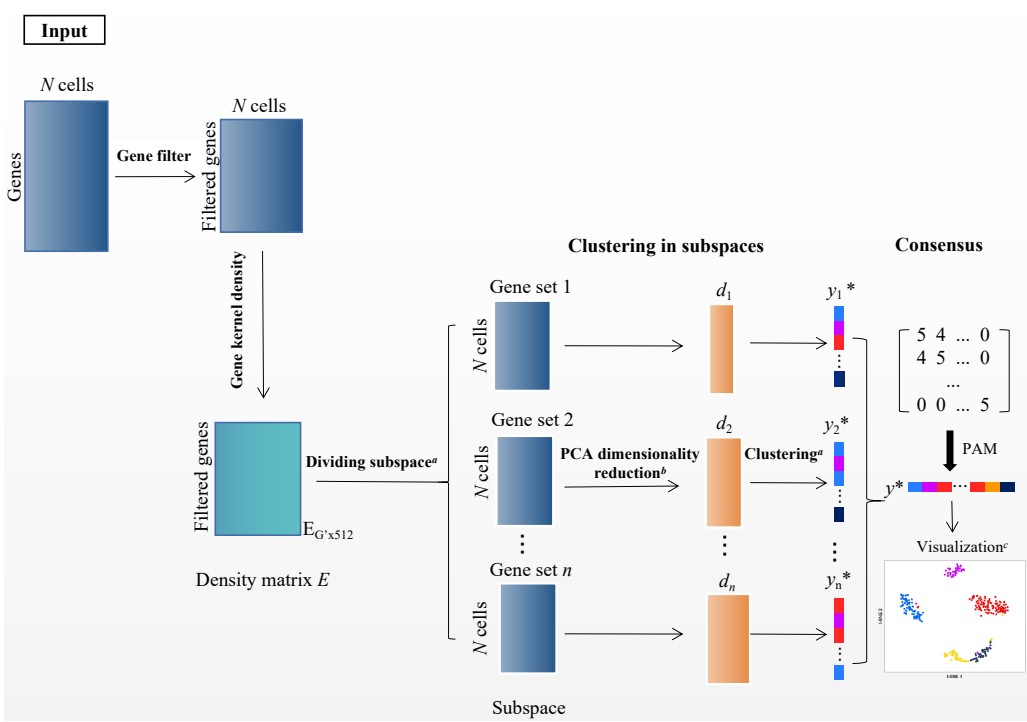

**Figure 2** **The SSWD framework for clustering scRNA-seq data.** (A) clustering by the improved $k$-means with *EP_dis* and RCH; (B) retaining the $d$-dimension with the elbow method; (C) visualization of Tian307 gene expression profile under clustering results; *: each element represents a cell, and different colors represent different clusters under clustering.

$k$-means clustering algorithm, which uses the median of data points rather than the mean and minimizes a sum of pairwise dissimilarities instead of a sum of squared Euclidean distance as the objective function (*Park & Jun, 2009*). The PAM is more robust to noise and outliers than $k$-means. Then, we used the *Sil* index (*Rousseeuw, 1987*) to estimate the cluster numbers (cell groups).

## Time complexity of SSWD

The main time-consuming step of SSWD is clustering by the improved $k$-means with *EP_dis* and RCH. In step 2 (see The overview of the SSWD), we used the improved $k$-means algorithm in the density matrix. We denoted $n$ represents the sample numbers, $m$ represents the feature numbers, $k$ represents the cluster numbers, $NC$ represents the range of cluster numbers, $l$ represents the iteration numbers to determine the cluster centers, and $w\_step$ represents the search step. Since the $k<<n$, $NC<<n$, the step 2 time complexity holds about O($lmn$). In step 3, each subspace would be performed the improved $k$-means algorithm after PCA. We denoted $d$ as the retained dimension after PCA, and $s$ is the number of genes subspace. The SSWD time complexity has roughly O($lmn + lnds$). Since $d<<m$, $s<<m$, we can simplify the time complexity of SSWD to approximately O($lmn$).

## Biological insights

We transformed the clustering results of each cell group into "one-against-the-rest". Then, we executed the Wilcoxon rank-sum test for each gene between the expression value and the binary cluster, adjusting the $p$-value based on FDR. The gene that adjusted $p$-value<0.001 was preserved as the ***differential gene***. Next, we used the AUC score to evaluate the performance of genes in distinguishing different cell types. Since AUC was only suitable for dichotomous problems, we constructed a binary classifier based on the mean expression value of each gene and compared the processed values with the binary cluster value. We defined the genes with AUC >0.85 and $p$-value<0.001 as ***marker genes***.

## Evaluation metrics

Two external validation indices, ARI (Adjusted Rand Index) and NMI (Normalized Mutual Information) were used to evaluate the effectiveness of clustering methods.

ARI (*Hubert & Arabie, 1985*) is a widely used external validation index in clustering, and it is defined as follows:

$$ARI(R,C) = \frac{\sum_{ij}\binom{n_{ij}}{2} - \left[\sum_i\binom{a_i}{2}\sum_j\binom{b_j}{2}\right]/\binom{n}{2}}{\frac{1}{2}\left[\sum_i\binom{a_i}{2} + \sum_j\binom{b_j}{2}\right] - \left[\sum_i\binom{a_i}{2}\sum_j\binom{b_j}{2}\right]/\binom{n}{2}}, \tag{4}$$

where $R$ and $C$ are published and predicted clusters, respectively. The overlap of samples between $R$ and $C$ can be generalized into a contingency table. $n_{ij}$ is the times a sample occurs in the $i$th cluster of $R$ and the $j$th cluster of $C$, $a_i$ is the sum of the $i$th row in the contingency table, $b_j$ is the sum of the $j$th column in the contingency table, and $(.)$ represents the binomial coefficient.

NMI (*Strehl & Ghosh, 2002*) is defined as follows:

$$NMI(R,C) = \frac{2*I(R,C)}{[H(R) + H(C)]}, \tag{5}$$

where $I(R,C) = \sum_{i=1}^{|R|}\sum_{j=1}^{|C|}p_{ij}\log(\frac{p_{ij}}{p_i p_j})$ is the mutual information between $R$ and $C$, $H(R) = -\sum_{i=1}^{|R|}p_i \log p_i$ is the entropy with $R$, $H(C) = -\sum_{i=1}^{|C|}p_i \log p_i$ is the entropy with $C$, $p_{ij} = \frac{n_{ij}}{n}$ is the probability that a cell belongs to both the $i$th cluster in $R$ and the $j$th cluster in $C$. The range of ARI and NMI are [0, 1]. The larger ARI (NMI) represent a better performance of clustering.

## Reference methods

In this article, seven prevailing clustering algorithms were introduced as reference methods. The SC3 v.1.22.0 (*Kiselev et al., 2017*), CIDR v.0.1.5 (*Lin, Troup & Ho, 2017*), Seurat v.4.1.1 (*Satija et al., 2015*) , SIMLR v.1.20.0 (*Wang et al., 2017*) were implemented with the original R package in Rstudio4.0. SinNLRR (*Zheng et al., 2019*) (https://github.com/zrq0123/SinNLRR) and S3C2 (*Zhuang et al., 2021*) (https://github.com/Cuily-v/S3C2) were implemented in Matlab2017a. The SNN-Cliq (*Xu & Su, 2015*) (https://github.com/BIOINSu/SNN-Cliq) was run in Matlab2017a and Python3.8.

SILMR and SNN-Cliq used the same log transformation as this paper. SinNLRR and S3C2 used the correct cell groups for clustering. Unless specified, the default parameters in the program were used as suggested in the original paper.

## RESULTS

### Performance evaluation and comparison with reference methods

We compared the performance of SSWD with seven prevailing clustering methods in eight scRNA-seq datasets (Table 4). The SSWD achieved the best clustering performance with an average ARI of 0.791 and was 0.143 higher than the second-ranked SC3, whereas the SNN-Cliq had poor performance (ARI of 0.364). SSWD ranked in the top three for ARI on all other datasets except Yan. SSWD attained the best results for NMI in three datasets (Li, Tian305, Tian307) and the second-best in four datasets (Biase, Yan, Deng, Treutlein). The average NMI of SSWD was the highest (0.850). Seurat had the poorest performance with only 0.579 in NMI because it failed on Biase, and the NMI of Li was only 0.122.

We further demonstrated the SSWD performance by ranking clustering accuracies on eight datasets (Fig. 3). For ARI (Fig. 3A), SSWD was superior to the seven reference methods in rank-wise (median of 2). SNN-Cliq performed the worst, with a median of 7. For NMI (Fig. 3B), SSWD was also superior to others, and the performance of CIDR, Seurat, and SNN-Cliq was all poor. Furthermore, the one-sided Wilcoxon signed-rank test was used to explain the statistical difference between SSWD and the reference methods. Except for SC3 and SinNRLL (in ARI), all the $p$-value are less than 0.05, which shows SSWD is superior to other methods (Table 5).

The SSWD was also better for estimating the cell groups. Five out of eight datasets (Biase, Patel, Li, Tian307, Tian305) acquired the correct cell groups using SSWD. Especially for Tian307 and Tian305, only the SSWD estimated the correct cell groups and achieved the best ARI of over 0.948. Deng contains seven cell groups, for which all methods failed to identify the correct number of cell groups. For Treutlein, CIDR and Seurat estimated the correct cell groups, but the ARI (0.188 and 0.531) and NMI (0.304 and 0.648) were lower than those of SSWD (0.607 and 0.732). For Yan, the SinNRLL and S3C2 performed very well under the correct cell groups.

### Annotate the clusters

We illustrated the effectiveness of the cell annotation using PanglaoDB (*Oscar, Li-Ming & Johan, 2019*) to clusters taking the Li dataset as an example. Li is a human pancreatic islet cells dataset containing five subtypes (alpha, beta, pp, acinar, and ductal) (*Li et al., 2016*). According to the AUC score (see Biological insights), we obtained the marker genes for each cluster identified by SSWD. Figure 4 is the expression heatmap of the top 10 marker genes for each cluster, which was divided into five clear modules and indicated that these marker genes could distinguish the clusters well. The *keration8* (*KRT8*) in cluster 1; *transthyretin* (*TTR*), *glucagon* (*GEG*) in cluster 2; *insulin* (*INS*) in cluster 3; *pancreatic polypeptide* (*PPY*) in cluster 4; *REG1B, REG1A, CTRB2* in cluster 5 were all reported in the original publication. We also annotated the cluster with PanglaoDB. The cluster

**Table 4 The performance of SSWD.** —The method that fails in clustering. () The actual cell groups have been provided as prior parameters. The best accuracy and the correct number of clusters (cell groups) are marked as bold for each dataset.

| Datasets | Actual cell groups | Measure | SSWD | SC3 | CIDR | Seurat | SIMLR | SNN-Cliq | SinNRLL | S3C2 |
|---|---|---|---|---|---|---|---|---|---|---|
| Biase | 3 | $k$ | **3** | **3** | 5 | — | 7 | 7 | (3) | (3) |
| | | ARI | 0.948 | 0.948 | 0.795 | — | 0.521 | 0.445 | **1.00** | 0.948 |
| | | NMI | 0.929 | 0.929 | 0.860 | – | 0.610 | 0.672 | **1.00** | 0.929 |
| Li | 5 | $k$ | **5** | 3 | 9 | 2 | 9 | 7 | (5) | (5) |
| | | ARI | **0.967** | 0.292 | 0.072 | 0.045 | 0.317 | 0.746 | 0.057 | 0.080 |
| | | NMI | **0.964** | 0.449 | 0.288 | 0.122 | 0.504 | 0.835 | 0.177 | 0.191 |
| Patel | 5 | $k$ | **5** | 18 | 7 | 6 | **5** | 26 | (5) | (5) |
| | | ARI | 0.776 | 0.445 | 0.744 | 0.689 | 0.809 | 0.278 | **0.849** | — |
| | | NMI | 0.762 | 0.668 | 0.846 | 0.680 | **0.849** | 0.463 | 0.823 | — |
| Deng | 7 | $k$ | 4 | 5 | 5 | 4 | 9 | 16 | (7) | (7) |
| | | ARI | 0.526 | **0.530** | 0.513 | 0.390 | 0.484 | 0.346 | 0.272 | 0.387 |
| | | NMI | 0.751 | 0.738 | 0.725 | 0.602 | **0.755** | 0.639 | 0.505 | 0.609 |
| Treutlein | 4 | $k$ | 6 | 7 | **4** | **4** | 10 | 19 | (4) | (4) |
| | | ARI | 0.607 | **0.724** | 0.188 | 0.531 | 0.353 | 0.209 | 0.583 | 0.475 |
| | | NMI | 0.732 | **0.850** | 0.304 | 0.648 | 0.534 | 0.505 | 0.664 | 0.644 |
| Yan | 7 | $k$ | 10 | 6 | 5 | 3 | 10 | 13 | (7) | (7) |
| | | ARI | 0.591 | 0.650 | 0.602 | 0.685 | 0.473 | 0.568 | **0.782** | 0.718 |
| | | NMI | 0.803 | 0.784 | 0.718 | 0.784 | 0.744 | 0.802 | 0.783 | **0.829** |
| Tian307 | 5 | $k$ | **5** | 7 | **5** | **5** | 8 | 42 | (5) | (5) |
| | | ARI | **0.958** | 0.745 | 0.651 | 0.910 | 0.576 | 0.154 | 0.915 | 0.955 |
| | | NMI | **0.945** | 0.836 | 0.714 | 0.885 | 0.733 | 0.546 | 0.888 | 0.938 |
| Tian305 | 5 | $k$ | **5** | 8 | 6 | 6 | 10 | 45 | (5) | (5) |
| | | ARI | **0.948** | 0.841 | 0.585 | 0.802 | 0.396 | 0.148 | 0.593 | 0.694 |
| | | NMI | **0.909** | 0.872 | 0.655 | 0.906 | 0.644 | 0.531 | 0.692 | 0.819 |
| Category correct ratio (%) | | | **62.5** | 12.5 | 25.0 | 25.0 | 12.5 | 0 | — | — |
| | Average | ARI | **0.791** | 0.647 | 0.519 | 0.507 | 0.491 | 0.364 | 0.631 | 0.608 |
| | | NMI | **0.850** | 0.766 | 0.639 | 0.579 | 0.672 | 0.624 | 0.692 | 0.709 |

results annotated with PanglaoDB are consistent with the cell annotations in the original publication (Table 6).

## DISCUSSION

### Role of the *EP_dis* metric

*EP_dis* was used in SSWD to assess the similarity between cells or genes. According to the *EP_dis* definition (see Materials & Methods), the optimal $w$ was determined by $SS_B/SS_W$ using a search strategy. When $w = 1$, the *EP_dis* equals the Euclidean distance; when $w = 0$, it is the Pearson distance. We used two simulated datasets, D1 and D2, to display the impact of *EP_dis* and explain the process of optimizing $w$ by $SS_B/SS_W$. Figure 5 shows the clustering accuracy of D1 (Fig. 5A) and D2 (Fig. 5B) under different $w$. It can be seen that the highest scores (*CA*, *Rand*, and $SS_B/SS_W$) in D1 and D2 are not appearing at the

Peer J

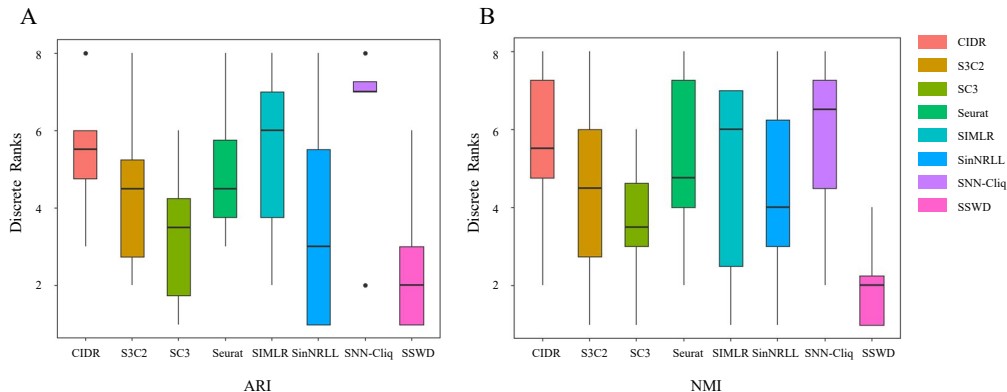

**Figure 3** The ranking performance of eight clustering methods on eight datasets. Each method is ranked according to ARI (A) and NMI (B) for eight datasets. A lower rank represents better performance (1 is the best and 8 is the worst). Ties are replaced by the mean of their ranks.

**Table 5** The results of the Wilcoxon signed-rank test conducted on SSWD versus the reference algorithms. The $p$-value ( $< 0.05$ ) indicates the significant difference between SSWD and the reference algorithms.

| Measure | SC3 | CIDR | Seurat | SIMLR | SNN-Cliq | SinNRLL | S3C2 |
|---------|------|------|--------|-------|----------|---------|------|
| *ARI* | 0.074 | 0.004 | 0.014 | 0.004 | 0.002 | 0.150 | 0.012 |
| *NMI* | 0.074 | 0.010 | 0.002 | 0.014 | 0.002 | 0.049 | 0.012 |

endpoints (0.6 in D1 and 0.8 in D2), which indicates that the *EP_dis* could capture more information between samples.

## Role of the relative CH

The RCH in the improved $k$-means algorithm was used to determine the cluster numbers. In SSWD, we employed RCH to estimate the gene subspace numbers and guide each set of gene subspace grouping. The capability of the RCH directly affects the performance of the SSWD. We utilized simulated datasets D3–D6 with different characteristics, six UCI datasets, and three scRNA-seq datasets to illustrate RCH properties and compare them with CH (Table 7). We can see that CH and RCH were consistent in D3–D6, indicating their good performance in simulated datasets. In the UCI datasets, RCH could estimate the correct cluster numbers except for Dermatology and Yeast, but the corresponding cluster numbers estimated by the RCH was closer to the real value than those of CH. For the scRNA-seq datasets, RCH and CH all failed. Their poor performance may be due to the characteristics of scRNA-seq data. Nonetheless, the RCH result was closer to the true value.

## Role of the subspace

After performing steps 1 and 2 of SSWD (see "Materials & Methods"), the Li has been separated into eight sets of gene subspace, and seven participate in consensus clustering (Fig. 6, Fig. S2). The expression heatmaps of the best three sets of genes subspace display clear patterns (Figs. 6A–6C), and their *EP_dis* heatmap (Figs. 6D–6F) effectively clustered

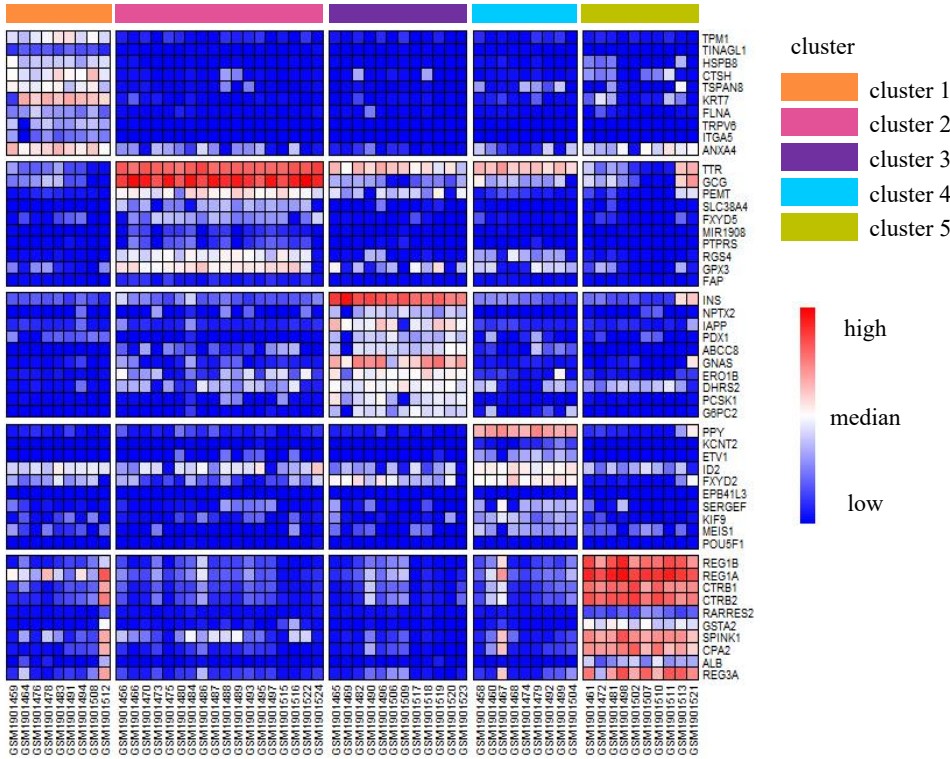

**Figure 4** **The expression heatmap of the top 10 marker genes for each cluster in Li.** Rows represent genes, columns represent cells.

cells with similar expression patterns. Compared with the *EP_dis* heatmap by 1,000 genes (Fig. 6G), the consensus matrix using sets of genes subspace (Fig. 6H) enhances intercellular signaling. The consensus matrix clustering result was better (ARI of 0.967, NMI of 0.964) than the former (ARI of 0.386, NMI of 0.579) because the former could not distinguish alpha and pp cells well.

## Discussion of prevailing methods

We also provided further discussion in Tables 4–5. The average performance of SC3 was only lower than SSWD, and its results were not significantly different from SSWD in the one-sided Wilcoxon signed-rank test. The SC3 combined multiple similarity measures (Euclidean, Pearson, Spearman) in clustering. It used the consistency matrix to integrate the multiple clustering results, and the consistency matrix strengthened the consensus signal between cells. At the same time, we can see that Deng and Treutlein, the best accuracy performers in SC3, could not obtain the correct number of cell groups. Although Biase estimated the correct number of cell groups in SC3, one cell was classified mistakenly, while SinNRLL could classify all cells accurately. Both SinNRLL and S3C2 introduce the idea of subspace clustering. Their average performances were better than other methods except for SSWD and SC3. However, this result was based on the cell group numbers being provided. Evaluating the cluster numbers is an important aspect of clustering methods.

**Table 6 Cluster annotation with the top marker genes and the PanglaoDB for the Li dataset.**

| SSWD results | Marker genes | AUROC | Adjust p-value | Cell type annotion with PanglaoDB |
|---|---|---|---|---|
| cluster 1 | | | | |
| | CTSH | 0.997 | 2.56E−06 | ductal cells |
| | KRT8 | 0.995 | 5.86E−06 | |
| | ANXA4 | 0.989 | 3.89E−06 | |
| cluster 2 | | | | |
| | TTR | 1.00 | 1.52E−09 | alpha cells |
| | GCG | 0.951 | 1.52E−09 | |
| | PEMT | 0.904 | 3.70E−08 | |
| | FXYD5 | 0.886 | 1.37E−07 | |
| cluster 3 | | | | |
| | INS | 1.00 | 1.23E−07 | beta cells |
| | NPTX2 | 0.946 | 1.67E−06 | |
| | IAPP | 0.933 | 4.69E−06 | |
| | PDX1 | 0.924 | 1.25E−06 | |
| | ERO1B | 0.917 | 3.83E−07 | |
| | PCSK1 | 0.889 | 1.67E−06 | |
| | G6PC2 | 0.886 | 3.13E−07 | |
| cluster 4 | | | | |
| | PPY | 1.00 | 2.31E−06 | pp cells |
| | ETV1 | 0.960 | 4.31E−06 | |
| | FXYD2 | 0.955 | 2.12E−05 | |
| | MEIS1 | 0.934 | 3.73E−05 | |
| cluster 5 | | | | |
| | REG1B | 1.00 | 8.22E−07 | acinar cells |
| | REG1A CTRB1 | 0.996 | 8.22E−07 | |
| | CTRB2 | 0.996 | 9.13E−07 | |
| | RARRES2 | 0.996 | 1.25E−06 | |
| | SPINK1 | 0.977 | 1.01E−06 | |
| | CPA2 | 0.977 | 1.25E−06 | |

Although SinNRLL could estimate the cluster numbers by other methods, its accuracy is still unsatisfactory (*Zheng et al., 2019*).

SNN-Cliq performed the worst (ARI = 0.364, NMI = 0.624), with none of the seven datasets estimating the correct number of cell groups. SNN-Cliq tended to divide more clusters, probably because the method requires providing three suitable parameters, and the results depend on the graphical representation of the data. CIDR used an implicit imputation approach to reduce the impact of dropout in scRNA-seq and used CH to estimate the cell groups. The method determined the correct number of cell groups in Treutlein and Tian307, but their clustering accuracies were poor. SIMLR adopts a

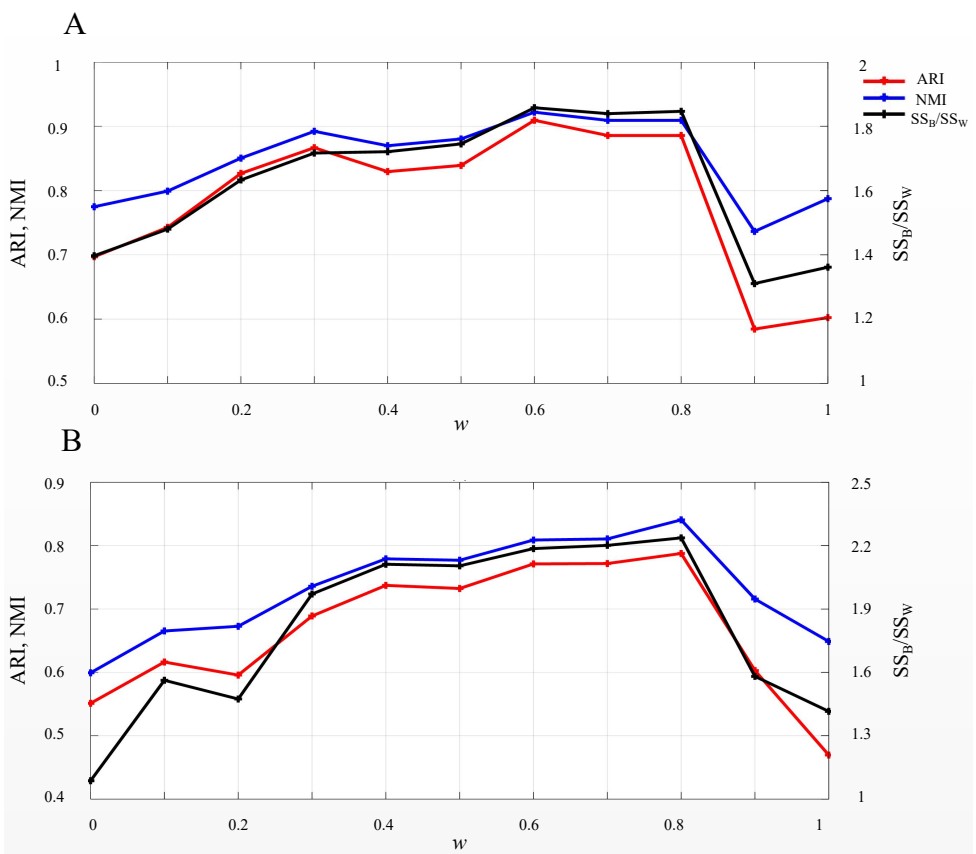

**Figure 5** $SS_B/SS_W$, *ARI*, and *NMI* values on D1 (A) and D2 (B) under different *w*. The left vertical axis in each subplot represents the values of *ARI* and *NMI* indices, and the right axis represents the $SS_B/SS_W$.

multi-kernel strategy to adaptively select an appropriate distance metric and automatically determine the cell groups. However, this method achieved good performance only in Patel because it used Euclidean distance as the metric to construct a Gaussian kernel function (*Taiyun et al., 2018*). For Seurat, Biase failed, and the ARI of Li was only 0.084. The results show that Seurat may be unsuitable for small datasets, consistent with the literature (*Kiselev, Andrew & Hemberg, 2019*).

The SSWD had the best performance in experiments. However, the performance of Patel and Yan were mediocre. Although Patel estimated the correct cell groups, the clustering accuracies were only ranked the third (in ARI) and the fourth (in NMI), probably because there were negative values in Patel datasets. All methods failed to estimate the correct cell group numbers in Yan. The poor performance of Yan in SSWD was because the estimated cell groups was far from the actual numbers.

We can draw the following conclusions from the above observations: (1) Due to the complex structure of scRNA-seq data, developing an optimal clustering method for all situations is impossible. (2) Determining the cluster numbers is difficult, so assigning cells to appropriate types is more important. (3) Selecting suitable similarity measures and using subspace in single-cell clustering help obtain better clustering results.

**Table 7  Comparison of the estimated cluster numbers between the CH and RCH under simulated and real datasets.** The correct number is marked as bold for each dataset.

| Datasets | True cluster number | Measure | |
|---|---|---|---|
| | | CH | RCH |
| D3 | 5 | **5** | **5** |
| D4 | 5 | **5** | **5** |
| D5 | 2 | **2** | **2** |
| D6 | 5 | **5** | **5** |
| Dermatology | 6 | 4 | 5 |
| Seed | 3 | 2 | **3** |
| Sensor | 4 | 2 | **4** |
| Statlog | 6 | 3 | **6** |
| Waveform | 3 | 2 | **3** |
| Yeast | 10 | 7 | 9 |
| Biase | 3 | 2 | 2 |
| Tian307 | 5 | 2 | 3 |
| Yan | 7 | 2 | 9 |

## CONCLUSIONS

The identification of cell types is a fundamental problem in scRNA-seq data analysis. In recent years, many clustering methods have been proposed. Most of them focus on computing more accurate and robust similarity measures between cells. However, conventional similarity measures are encountering challenges to single-cell data clustering because of the high dimensional, high noise, and high dropout. This study proposed a clustering method for small scRNA-seq data, named as SSWD, based on sets of gene subspace and weighted distance. Firstly, an improved $k$-means with $EP\_dis$ and RCH was applied to divide sets of gene subspace with similar density distributions, which better identify distinct cell groups. Secondly, cell clustering was performed in these sets of gene subspace. Lastly, the ensemble clustering with PAM was conducted on the consensus matrix composed of gene subspace clustering results. The results of eight scRNA-seq datasets showed that SSWD could effectively reduce the influence of noise in clustering and better capture the intrinsic relationship between cells or genes, thereby achieving more robust and accurate clustering results.

### Funding

This work was supported by the Natural Science Foundation of Hunan Province (2021JJ30351), the Scientific Research Project of Hunan Provincial Department of Education (21B0187), and the Special Funds for Construction of Innovative Provinces in Hunan Province (2021NK1011). The funders had no role in study design, data collection and analysis, decision to publish, or preparation of the manuscript.

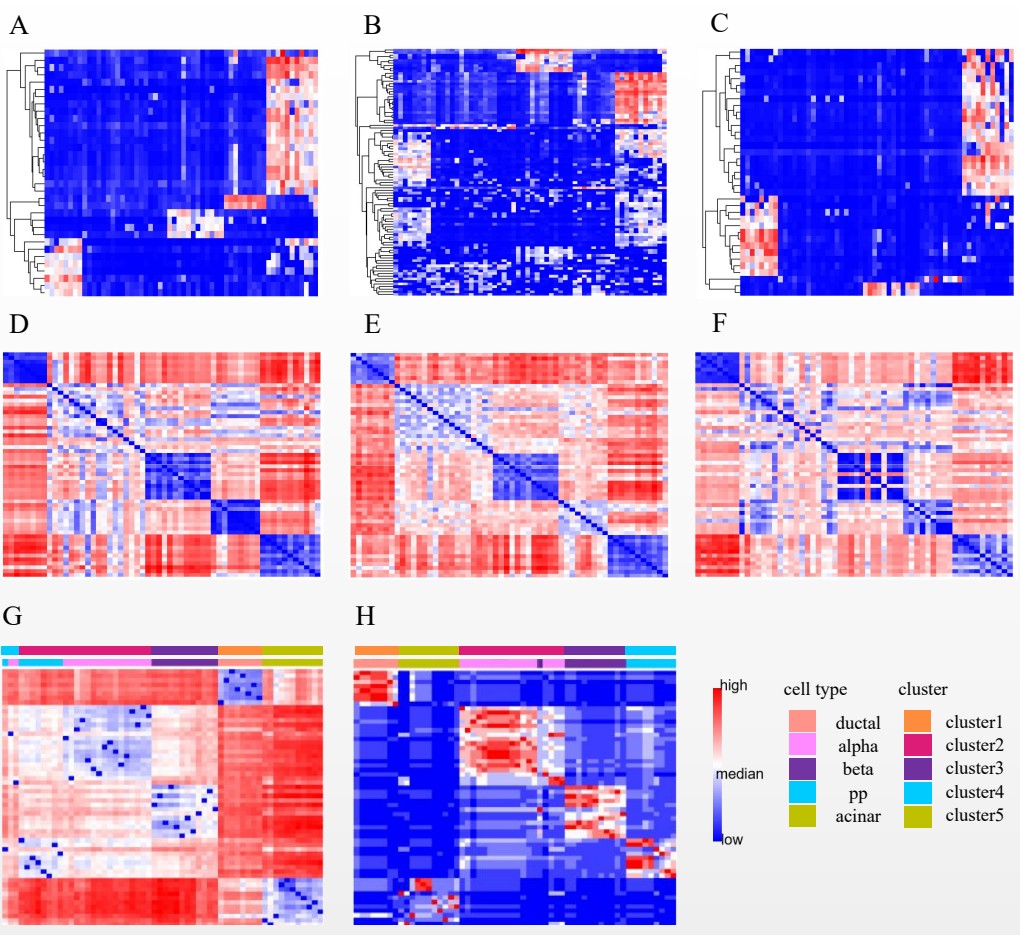

**Figure 6 Three subspace expression heatmaps and distance heatmaps for the LI dataset.** (A–C) are the best three subspaces expression heatmap; rows represent genes, columns represent cells; (D–F) are the *EP_dis* distance heatmap relating to (A–C); (G) is the *EP_dis* distance heatmap with 1,000 genes; (H) is the consensus matrix heatmap; in (D–H), both rows and columns represent cells. In (G) and (H), the first color bar is the cell groups after clustering by SSWD, and the second is the actual cell types.

## Grant Disclosures

The following grant information was disclosed by the authors:
Natural Science Foundation of Hunan Province: 2021JJ30351.
Scientific Research Project of Hunan Provincial Department of Education: 21B0187.
Special Funds for Construction of Innovative Provinces in Hunan Province: 2021NK1011.

## Competing Interests

The authors declare there are no competing interests.

## Author Contributions

- Zilan Ning conceived and designed the experiments, performed the experiments, analyzed the data, prepared figures and/or tables, authored or reviewed drafts of the article, and approved the final draft.

- Zhijun Dai performed the experiments, authored or reviewed drafts of the article, and approved the final draft.
- Hongyan Zhang analyzed the data, prepared figures and/or tables, and approved the final draft.
- Yuan Chen analyzed the data, prepared figures and/or tables, authored or reviewed drafts of the article, and approved the final draft.
- Zheming Yuan conceived and designed the experiments, authored or reviewed drafts of the article, and approved the final draft.

## Data Availability

The raw data is available at NCBI GEO: GSE57249, GSE73727, GSE57872, GSE45719, GSE52583, GSE118767, GSE36552.

The code of SSWD described in this article is available at Github: https://github.com/ningzilan/SSWD; Zilan Ning. (2022). SSWD: The clustering method for small scRNA-seq data based on subspace and weighted distance. Zenodo. https://doi.org/10.5281/zenodo.7471227.

## Supplemental Information

Supplemental information for this article can be found online at http://dx.doi.org/10.7717/peerj.14706#supplemental-information.

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
