# Peer review of "A clustering method for small scRNA-seq data based on subspace and weighted distance"

_PeerJ, doi:10.7717/peerj.14706_

## Round 0.1 · original submission · Minor Revisions

As the authors will realize, the content of the manuscript needs to be improved per the reviewers' comments. Therefore, I would like to invite the authors to respond to the reviewers' comments as well as to implement the suggestions where appropriate.

Reviewer 1 ·

Basic reporting

no comment.

Experimental design

no comment.

Validity of the findings

The performance of the newly developed method (SSWD) was compared with seven well-known clustering methods in eight datasets by the authors. Their method, SSWD, performed better than other methods with the greatest ARIs only in three datasets while having the best clustering performance when the average of ARI was taken into account. As there is no gold standard clustering approach in the literature and the clustering performance varies from data to data, I believe it is still worth publishing this study.

Additional comments

The authors tested the performance of their approach (SSWD) against seven well-known clustering techniques on eight datasets. Only in three datasets did their technique, SSWD, outperform other methods with the highest ARIs while having the best clustering performance when the average ARI was taken into account. These datasets, on the other hand, are made up of a very small number of cell groups. When there are many cell groups, such as at least 15 cell groups, how does the performance of SSWD change? This could be investigated by simulating scRNA-seq datasets using the "Splatter" package.

The datasets included in the study all have distinct unit counts. Was there a reason for this, according to the authors? If so, could you please explain the rationale? Otherwise, what would the results be if the same unit was used?

What is the time complexity of SSWD?

Did the authors create an R package for their clustering approach? Is there any link that the authors can give in the paper so that others can follow all the source codes and generate the same results?

Last but not least, although it is beyond the scope of the current work, I believe that employing various distance metrics, particularly Kullback-Leibler Divergence or Mahalanobis distance, in the same approach (SSWD) may improve performance even more.

Reviewer 2 ·

Basic reporting

The manuscript is laid out well, however, the language can be improved to correct grammar and syntax.
The research question is stated well. There is a need to describe several points, possibly with addition of a few references. All the necessary data is shared.

Experimental design

The authors define a heuristic method SSWD for clustering small scRNA-seq datasets. The performance is benchmarked along with 7 existing methods. On average, SSWD shows improved performance compared to any other of the 7 methods. There are multiple minor points that need to be addressed before this work could be published.

Validity of the findings

Statistical methods are adequate.

Additional comments

The most important points:

Title states that SSWD is an analysis pipeline, however SSWD is a clustering method designed and tested on small scRNA-seq datasets. The fact that the used datasets are small (49-430 cells) should be clearly stated.

Line 169-170: "to obtain genes'’ density profile". For the density function the input vector has to be ordered in some way. When you compute density for a gived genes how do you order cells? The matlab source code hints that the order is arbitrary. If that is the case, how your results change if you shuffle cells in the datasets before SSWD?

Line 29-32: "The EP_dis could simultaneously capture information reflecting the globally spatial correlation and locally variable trend in high-dimensional feature space by combining Euclidean distance and Pearson distance through a weighting strategy."
Provide references and arguments how Euclidean and Pearson distance would capture global correlation and locally variable trends in data. Otherwise, remove phrases "globally spatial correlation and locally variable" from the manuscript.
In the introduction comment on your choice of Euclidean and Pearson distances. There are several other popular metrics reviewed in https://academic.oup.com/bib/article/20/6/2316/5077112. Can other metrics be used with scRNA-seq data with SSWD? If so, how?

On line 79 you introduce term "density". Define the term. Describe the coordinate along which the density is calculated.

On line 84 you introduce term "consensus clustering process, which intensifies the consensus signal from multi-subspace". Clarify the procedure.

Line 125: "E and P must be normalized before calculating". Describe how this normalization is done.

Line 182: Define "PAM" and explain why is it chosen for clustering of matrix M.

Line 253: "... the five clusters consistent with the types provided in the original published". Authors are establihing validation of cell type annotation done in the Li paper. Pealse substantiate and argument how the markers shown in Table 4 are consistent with the Li paper.

Section "Role of the relative CH" uses "six real datasets" to demonstrate that RCH is better than CH. The type of these 6 datasets is different from scRNA-seq data. Comment why it is necessary to use the 6 datasets instead of scRNA-seq datasets.

Figure 1: What is the criterion to decide whether "The cluster centroid needs to be updated"? In the previous step there was already an action "Update cluster centroid".

Figure 2: Elaborate the caption.
In the figure and in the main text, indicate that PCA on z-score normalized data is done for each subspace separately followed by clustering.
In the figure indicate which clustering method is used whenever "Clustering" is done.
The last panel in the figure appears to be 2D projection of a dataset; please state clearly what it is.

Table 4: Are the cell type annotations in this table done by Li or with PanglaoDB tool? Also, specify the tool name. Add another column to have both annotations, e.g. "Cell type, Li et al." and "Cell type".

For each tested method specify exact versions of the software that were used.

In the methods, describe how the datasets D1 and D2 were generated.

Table 1: Add a column with the sequencing technology, e.g. Smart-seq, 10x etc.




Less crucial points:

Use of "spatial" with scRNA-seq data is misleading since scRNA-seq data does not have spatial caomponent and should be removed from the abstract and text.

Line 45 Word "efficient" need to be replaced with "widely used" or similar phase.

Line 48. Please comment on "high redunduncy" problem of the scRNA-seq data.

On line 61 "However, it is not scalable" is umbiguous what "it is" was referred to. Please clarify.

Line 67. It might be necessary to start a new paragraph.

Line 75. Sentence "But all of them have to provide the number of cell groups as prior parameters." is not accurate and needs to be removed.

Line 76. Word "inspired" need to be replaced with "highlighted" for clarity.

Clearly define the terms "subspace" and "multi-subspace", e.g. a set of genes, sets of genes, etc.

Line 142: "N" in the equation 2 should read "NC"

Line 177: Define term "signal"

Line 191: Clarify what "gene prediction ability" exactly means.

Line 270: Text reads "two-dimensional Gaussian datasets" but the Table 5 caption states 30 and 10-dimensional data. Resolve the inconsistency.

Line 277: "The expression heatmap of the best three subspaces ...". What is the criterion to select the 3 best subspaces?

Figure 3: Caption (and the main text as well) states "seven clustering methods". There are 8 methods in the figure.

Figure 5: Update left axes labels to read "ARI, NMI" since currently it seems to be a ratio of ARI/NMI.

Figure 6: Add x and y axes labels.

Table 1: The header states "Genes", replace it with variables since some of the entries have genes and other have transcripts. Clarify this in the caption.
Spell out "Reference".

Table 5: What is the distribution that the random error is drawn from?

Define UCI in text and tables.

The source code contains comments in non-English characters; translate them with English.

Reviewer 3 ·

Basic reporting

The level of English is satisfactory. There are small grammatical mistakes in the manuscript, so it needs a prof-reading.

Literature references are sufficient in terms of background coverage.

Resolutions of figures are generally acceptable, due to embedding into .pdf file, some fonts became blur, they would be corrected in revision.

Experimental design

The manuscript developed a new pipeline for ScRNA-seq data and designed a new distance metric to capture spatial correlation and variable trends among cells/genes. They experimented the SSWD on eight public scRNA-seq datasets and compared it with seven scRNA-seq clustering methods. Experimental design and application of methods are acceptable.

Authors should explain some questions:

1. What was the selection criteria for eight public scRNA-seq datasets?

2. To perform cell annotation of clusters, they chose Li dataset (line 249)? Why was this dataset chosen for annotation purpose?

3. Based on results given in Table 4, are clusters’ content or marker genes the same with the referred publication results?

Validity of the findings

Authors should reply question and clarify some unclear points in the manuscript.

1. In line 280, authors wrote that "the consensus matrix using subspace (Fig. 7G) enhances intracellular signaling.” How did authors observe an improved intracellular signaling in Fig. 7G?

2. The discussion section should be extended; it is very short based on number of experiments. There should be more discussion about prevailing methods because there is very limited discussion about pros-cons of these methods.

3. Authors should clarify why SSWD performed worse than some prevailing methods in some datasets.

Additional comments

I do not have any other comment.

---

## Round 0.2 · Major Revisions

Some part of the manuscript shows substantial similarity to Song et al. 2021. Therefore, I would like to invite the authors for a more detailed check to ensure the manuscript is not plagiarizing any published works.

Reviewer 1 ·

Basic reporting

no comment

Experimental design

no comment

Validity of the findings

no comment

Additional comments

I have gone through the track-changes manuscript and rebuttal letter and see that the authors addressed my concerns and substantially improved the content of the manuscript. So, the manuscript may be now accepted for publication in its current form.

Reviewer 2 ·

Basic reporting

The authors have addressed all my comments and questions.

Experimental design

The authors have addressed all my comments and questions.

Validity of the findings

The authors have addressed all my comments and questions.

Additional comments

The authors have addressed all my comments and questions.

Reviewer 3 ·

Basic reporting

The level of English is satisfactory.

Literature references are sufficient in terms of background coverage.

Resolutions of figures are acceptable.

Experimental design

The manuscript developed a new pipeline for ScRNA-seq data and designed a new distance metric to capture spatial correlation and variable trends among cells/genes. They experimented the SSWD on eight public scRNA-seq datasets and compared it with seven scRNA-seq clustering methods. Experimental design and application of methods are acceptable.

We will not ask any further question about the revised manuscript.

Validity of the findings

Authors replied my questions and made necessary updates in the manuscript.

Additional comments

I do not have any other comment.

---

## Round 0.3 · accepted · Accept

The authors have satisfactorily addressed the concerns during the review process. The manuscript can now be accepted for publication.